# OpenReview forum: "TruthFlow: Truthful LLM Generation via Representation Flow Correction"
_ICML.cc/2025/Conference — ICML 2025 poster_

### Official Review · Reviewer_EcU5 · 2025-03-05

**Overall Recommendation:** 3

**Summary:**

The paper  introduces TruthFlow, a method that enhances LLM truthfulness by learning query-specific correction vectors via Flow Matching. Unlike prior universal intervention approaches, TruthFlow generates corrections for each query specifically that transition representations from hallucinated to truthful states. It applies these corrections at specific transformer layers and refines them via subspace projection. Experiments on TruthfulQA show significant truthfulness improvements over baselines like ITI and TruthX, with better  generalization to unseen benchmarks. Ablation studies confirm the effectiveness of query-specific correction and subspace filtering in mitigating hallucinations.

**Claims And Evidence:**

The claims in the paper are generally well-supported by empirical evidence, but some aspects could be strengthened:

- The paper claims that universal correction vectors are insufficient for truthfulness correction, because the direction is dependent on query. It is supported by PCA visualizations and experimental results showing superior performance of TruthFlow over other methods with universal correction. However, the visualization in Figure 2 only provides qualitative evidence, and additional statistical analyses on vector directions distributions would strengthen the argument.
- The paper demonstrates results on HaluEval, Natural Questions, and TriviaQA, suggesting that TruthFlow generalizes well. While the results are better than other methods, the quality drop is significant. I believe it would be better to show the full cross-domain performance matrix. Also further validation across diverse domains (e.g., medical or legal datasets) would enhance this claim, if possible. It is also not entirely clear why the hyperparameters are different for different datasets, if TruthFlow was trained only on TruthfulQA.

**Essential References Not Discussed:**

I believe the paper cites and discusses the relevant work necessary to understand its key contributions, and I do not see any essential missing references.

**Experimental Designs Or Analyses:**

The evaluation on TruthfulQA and transferability tests on HaluEval, NQ, and TriviaQA are appropriate for measuring truthfulness improvements. However, it is not clear why alpha are different for these datasets, which probably leads to unfair comparison. The use of GPT-4-based scoring, BLEURT, and multiple-choice accuracy metrics makes sense but introduces potential biases (e.g., LLM-based evaluation may not always be reliable).

**Methods And Evaluation Criteria:**

Yes, the proposed methods and evaluation criteria are generally well-aligned with the problem of improving truthfulness in LLMs.

**Other Comments Or Suggestions:**

I have one more small suggestion. It would be very interesting to see failure cases for the TruthFlow. For instance, does the method sometimes overcorrect and reduce informativeness? Addressing this would provide a deeper understanding.

Also typo in lines 367-368 "Natrual Questions"

**Other Strengths And Weaknesses:**

Strengths:
- **(S1)** Novelty of query-specific approach: the paper introduces a novel query-specific approach to representation intervention, addressing the limitations of universal correction vectors used in prior methods like ITI. I believe that the correction based on specific query is a very promising direction. The application of Flow Matching for truthfulness correction is a fresh and useful perspective in this domain.
- **(S2)** Clarity: The paper is well-structured, with clear explanations of methodology, ablation studies, and evaluations. Almost all necessary details (e.g. hyperparams, architectures) could be found in the paper.
- **(S3)** Extensive ablations: The paper provides ablation studies almost on all topics like effect of layers, number of chosen singular vectors etc. It strengthens the paper and makes it more valuable

Weaknesses:
- **(W1)** Insufficient Statistical Evidence for Universal Correction Vector Limitation: the paper argues that universal correction vectors are inadequate because truthfulness correction direction depends on the query. This claim is supported by PCA visualizations and experimental results showing that TruthFlow outperforms universal correction methods. However, Figure 2 only provides qualitative evidence, and additional statistical analysis on vector direction distributions would strengthen this argument.
- **(W2)** Cross-Domain setup: while TruthFlow shows promising generalization on HaluEval, Natural Questions, and TriviaQA, the performance drop across domains is significant (related to training dataset). Also there is just a slight improvement upon the base model. A more detailed cross-domain performance matrix would provide better insights into where TruthFlow succeeds and where it struggles. Additionally, further validation on more diverse, real-world datasets would enhance the claim of strong generalization.
- **(W3)** Unclear Selection of Alpha Parameter: the choice of alpha (intervention strength) across different datasets is not well explained. It is unclear how this hyperparameter should be selected and why different values are used across datasets. This could lead to an unfair comparison if some baselines were not similarly fine-tuned. A more systematic explanation or tuning strategy for alpha would improve the reproducibility and fairness of the results.

**Questions For Authors:**

- **(Q1)**  In Figure 2, you provide a PCA visualization to support the claim that universal correction vectors are insufficient. Have you considered performing statistical analyses (e.g., distributional comparisons, cosine similarity metrics) to further validate the diversity of truthfulness correction directions across queries?
- **(Q2)** Your transferability experiments on HaluEval, Natural Questions, and TriviaQA show that TruthFlow generalizes better than baselines, but the quality drop is significant. Also there is just a slight improvement upon the base model. Could you provide a full cross-domain performance matrix to give a clearer picture of how this method perform across all datasets while transferring? Additionally, would it be possible to evaluate your method on other datasets to further assess real-world generalization?
- **(Q3)** The alpha parameter appears to vary across datasets, but the paper does not clearly explain how it is chosen. Could you clarify the selection process for alpha? Was it optimized separately for each dataset, and if so, how do you ensure a fair comparison with baselines that might not have been similarly fine-tuned? Would a fixed or adaptive alpha be a possible alternative to improve consistency?

**Relation To Broader Scientific Literature:**

The paper extends prior work on representation intervention (e.g., ITI, TruthX) and Flow Matching to improve LLM truthfulness. Unlike ITI, which applies a fixed correction vector, TruthFlow learns query-specific correction vectors, addressing the limitation of one-size-fits-all interventions.

**Theoretical Claims:**

The paper primarily focuses on empirical results rather than theoretical claims. No explicit errors were found in the provided equations.

---

> ### Author Rebuttal · Authors · 2025-04-01
>
> We thank the reviewer for their valuable feedback and constructive suggestions.
>
> ---
> **Q1: Statistical evidence**
> A1: Thank you for your suggestion. We further conduct the following statistical analysis to demonstrate this limitation. Specifically, we calculate the cosine similarity between the universal vector $y$ and each specific truthful correction vector $x_i$ obtained from LLM internal states. We first calculate the variance of the cosine similarities $\frac{\langle x_i, y\rangle}{\left\|x_i\right\|_2\left\|y\right\|_2}$ and find it to be 0.536. Considering the range of cosine similarity, this suggests a quite high statistical variance. Furthermore, we also visualize the distribution of the cosine similarities in the following URL to show the truthful correction vector diversity. https://anonymous.4open.science/r/13040_Rebuttal-432A/
>
> **Q2: Transferability** A2: Thanks for your suggestion.
> - First, we emphasize that there's no quality drop in our transfer experiment (only the improvement is less significant). This is perfectly normal as transferring across different datasets is indeed difficult and our method has already achieved better performance compared with baselines.
> - We understand that the reviewer asks for a clearer picture of the transfer performance. However, the full transfer performance matrix may not be a good idea and is not adopted in prior related works. The reason is that for data-driven methods such as TruthFlow, the transfer performance not only depends on the method itself but also on the quality of training data. Specifically, we find TruthfulQA more suitable for eliciting truthful answers because of the elaborate "traps" in the questions. The truthful correction vectors obtained under such a scenario may capture more truthful-related information. In contrast, questions in HaluEval, NQ, and TriviaQA are more related to specific knowledge. If the LLM lacks sufficient knowledge, the representations of correct and incorrect answers won't constitute a strong contrast in truthfulness. As pointed out by Reviewer P7qz, existing work [1] has shown that even if directly training intervention methods on datasets such as NQ, it may lead to incremental performance gain, or even drop compared to the base model, not to mention the transfer case.
> - To provide more evidence for transferability, we transfer TruthFlow to MedHallu [2]. The results show that TruthFlow can also be generalized to mitigate medical hallucinations by achieving improved performance compared to the base LLM.
>
> |Method|True|Info|True\*Info|
> | --- |:---:|:---:|:---:|
> |Llama3 base|45.8|91.8|42.04|
> |TruthFlow|46.5|94.1|43.76|
>
> [1] Liu et al., Enhancing Language Model Factuality via Activation-Based Confidence Calibration and Guided Decoding. EMNLP 2024
>
> [2] Pandit et al., MedHallu: A Comprehensive Benchmark for Detecting Medical Hallucinations in Large Language Models. arXiv preprint
>
> **Q3: Evaluation on other datasets for real-world generalization** A3: To test real-world generalization, we conduct experiments on a text summarization and a medical hallucination QA task. Please see our reply to Reviewer P7qz's Q3. The performance gains indicate that TruthFlow can generalize to more practical tasks. Besides, to check the general utility, we evaluate TruthFlow on MMLU. Please see our reply to Reviewer 1fpP's Q1. We observe a minor decrease which shows that TruthFlow doesn't hurt general utility while improving factuality.
>
> **Q4: Selection of $\alpha$.**
> A4: We apologize for the confusion.
> - In practice, we reserve a small part of the training set for hyperparameter tuning. Specifically, we conduct a grid search on $\alpha$ from $[0.5, 4.5]$ and pick the optimal one on the validation set.
> - To ensure a fair comparison, we conduct similar hyperparameter tuning on other baselines. For example, on ITI and NL-ITI, we conduct a grid search on the intervention intensity from $[1, 25]$. On AD, we tune the info layer ranging from 22 to 30 for 32-layer LLM and 28 to 38 for 40-layer LLM and tune the entropy control penalty coefficient $\lambda$ from $[0.4, 1]$.
> - Note that for transferability, since the flow model is trained on the original dataset, using the same $\alpha$ for other datasets is usually too large and can easily get overfitted. Thus our hope is to keep the ratio of the query representation norm and correction vector norm in a stable position. Specifically, we calculate the average query representation norm as $n_q$ and the average correction vector norm as $n_c$ and set $\alpha = \beta* n_q/ n_c$ where $\beta$ is a constant parameter (e.g., 0.1) and rounded to the nearest half-integer values.
>
> **Q5: Typos and failure cases**
> A5: Thanks for pointing out. We will fix the typos in the revision. Due to space constraints, we show some failure cases in https://anonymous.4open.science/r/13040_Rebuttal-432A/. We observe that some failures arise from the lack of necessary knowledge, which is consistent with the analysis in [1].

---

> > ### Comment · Reviewer_EcU5 · 2025-04-05
> >
> > I appreciate your revisions and responses. You've resolved the main issues I highlighted, and I’ve raised my score.

---

> > > ### Author Response · Authors · 2025-04-05
> > >
> > > Thank you for your positive feedback and thoughtful questions. We sincerely appreciate your recognition of our work and the improved score. Your comments have helped us strengthen the paper, and we will revise it accordingly to make it more complete and rigorous.
> > >
> > > Thank you again for your time and effort in reviewing our submission.

---

### Official Review · Reviewer_XSKe · 2025-03-15

**Overall Recommendation:** 4

**Summary:**

This paper introduces a novel method called TruthFlow, which enhances the ability of LLMs to generate truthful responses through representation flow correction. TruthFlow leverages flow matching techniques to generate query-specific truth-aligned correction vectors, guiding the model from a hallucinatory state to a truthful state. Experimental results on the TruthfulQA dataset demonstrate that TruthFlow reduces hallucinations and exhibits transferability across multiple datasets.

**Claims And Evidence:**

Yes.

**Essential References Not Discussed:**

NA

**Experimental Designs Or Analyses:**

See weaknesses.

**Methods And Evaluation Criteria:**

Yes.

**Other Comments Or Suggestions:**

I have no other comments or suggestions.

**Other Strengths And Weaknesses:**

## Strengths
1. The use of the Flow Matching technique for query-specific correction vectors is innovative and promising.
2. The paper provides evidence that the assumption of a Universal Correction Vector, relied upon by previous representation intervention methods, does not entirely hold in practice. This insight is clearly demonstrated through effective visualizations.
3. The experiments are comprehensive. On the TruthfulQA dataset, TruthFlow not only enhances truthfulness but also demonstrates generalizability across multiple datasets.

## Weaknesses
1. In Table 1, there are instances where the Info score is relatively low despite the True score being the highest. This may suggest that while TruthFlow improves the truthfulness of responses, it might also lead the model to provide more conservative answers.
2. The paper primarily evaluates on the TruthfulQA dataset, with transferability assessments conducted on common QA datasets. In the future, it would be beneficial to explore additional domains, such as legal or medical fields, to further demonstrate the universality and robustness of TruthFlow.

**Questions For Authors:**

NA

**Relation To Broader Scientific Literature:**

NA

**Theoretical Claims:**

Yes, I have checked the correctness of the theoretical claims.

---

> ### Author Rebuttal · Authors · 2025-04-01
>
> We thank the reviewer for their valuable feedback and constructive suggestions. We address the questions as follows:
>
> ---
> **Q1:Slightly lower Info score.**
>
> A1: We actually observed and analyzed this phenomenon in the paper's “Qualitative Study” paragraph before section 5. Specifically, we find that some of the best answers in TruthfulQA are labeled as less informative, (e.g. the best answer is “I have no comment”). And since TruthFlow makes the model output more truthful, it inevitably causes the Info score to be lower on those questions (see an example in Table 2). This leads to the overall Info score being lower. Please note that TruthFlow can still provide informative answers to other questions whose correct answer is indeed informative.
>
> **Q2: Additional domains, such as legal or medical fields.**
>
> A2: Thanks for your suggestion. We conduct additional experiments to apply TruthFlow on the medical QA task. Specifically, We test on MedHallu [1] dataset which contains 1000 human labeled medical questions, along with knowledge, ground truth, and hallucinated answer. Due to time constraints, we only compare TruthFlow with the base LLM and ITI. We follow the evaluation metric in TruthfulQA to calculate the True score, Info score, and True\*Info score using GPT-4o. The results are listed below. We can observe that TruthFlow still achieves significant improvement over the base model and ITI despite a slight decrease in the Info score.
>
> | Method      | True  | Info  | True\*Info |
> | ----------- |:-----:|:-----:|:----------:|
> | Llama3 base | 42.54 | 96.82 |   41.19    |
> | ITI         | 54.77 | 68.70 |   37.63    |
> | TruthFlow   | 57.21 | 94.87 |   54.27    |
>
> [1] Pandit et al., MedHallu: A Comprehensive Benchmark for Detecting Medical Hallucinations in Large Language Models. arXiv preprint

---

> > ### Comment · Reviewer_XSKe · 2025-04-08
> >
> > Thanks to the author for the reply, which have addressed my concerns. I will raise my score.

---

> > > ### Author Response · Authors · 2025-04-08
> > >
> > > Thank you for recognizing our work and for raising the score. We sincerely appreciate your time and effort in reviewing our paper.

---

### Official Review · Reviewer_1fpP · 2025-03-15

**Overall Recommendation:** 4

**Summary:**

In order to address the hallucination problem for LLMs, a line of methods, named representation intervention, attempts to edit LLMs' hidden representations at certain layers to guide their behavior, such as making the generated outputs more truthful. However, these methods usually assume that there exists some universal truthful intervention vector in the representation space of LLMs that turns any input query from its hallucinated states to the truthful states. In this paper, the authors show that such an assumption may not hold. Inspired, they propose a flow-matching-based representation intervention method which uses a flow matching model to learn query-specific correction vectors. Specifically, this flow matching model takes any specific query’s representations as input and output its corresponding truthful representation correction vector. Empirically, they demonstrate the effectiveness of the proposed model on TruthQA benchmark using various base models, compared with a comprehensive collection of baseline methods.

## Update after rebuttal
I raise my score since the author's responses have addressed my questions.

**Claims And Evidence:**

All the claims made in the submission are supported by clear and convincing evidence.

**Essential References Not Discussed:**

None.

**Experimental Designs Or Analyses:**

In this paper, the effectiveness of the proposed method in generating truthful and informative outputs is validated by the true/info scores on TruthfulQA, HaluEval, NQ, and TriviaQA datasets.

But, one typical concern in representation intervention is whether intervening on one direction, like truthfulness, will compromise the LLM's general utility. For example, the intervened LLM may generate very truthful text, but the fluency within the text could be compromised. Therefore, to know whether some core capabilities of LLMs are compromised, it would be good to test the intervened models on some related and fundamental benchmarks like ARC, HellaSwag, or MMLU.

**Methods And Evaluation Criteria:**

The datasets used in this paper, including TruthfulQA, HaluEval, NQ, and TriviaQA are all standard and popular datasets used to evaluate the truthfulness of LLMs.

**Other Comments Or Suggestions:**

None.

**Other Strengths And Weaknesses:**

1. The proposed method is well-motivated. The motivation introduced in Section 3.2, Figure 2 in particular, clearly shows why the assumption of universal intervention vectors may not hold. This is because some query-specific directions may contradict with the general direction.
2. This paper is well-written. The preliminaries are clear. It's easy to follow up the idea and understand the details throughout the paper.
3. The experiments are extensive, covering various base models and truthfulness datasets.

**Questions For Authors:**

1. Could you specify how to get the blue arrow (not the light blue arrow) in Figure 2?

2. As introduced in the related work section, there are different types of methods to improve LLMs' truthfulness, such as representation intervention, post-training, and contrastive decoding. It seems in the experiments, the baselines for post-training are not covered. If it is possible to compare the proposed method with existing post-training baselines? e.g., [1]

3. How important is the Truthfulness-Related Subspace Projection step? I understand that in Section 5.3, Figure 3 in particular, you've shown the performance comparison on different numbers of top singular vectors. But, it seems there is no comparison between using and not using the projection. So, I am wondering how important this projection step is. Moreover, according to Figure 3, it seems increasing the number of top singular vectors won't affect the performance? because the true/info scores seem similar across k=10,15,20,25.

4. I am just curious of the possibility of the following: It seems that the only "tool" you need is a generative model that can generate the target distribution from the source distribution. So if it is possible to replace the flow-matching model with some other types of generative models like the diffusion models?

[1] Chen, W., Song, D., and Li, B. Grath: Gradual selftruthifying for large language models. ICML, 2024.

**Relation To Broader Scientific Literature:**

This paper provides a good example of how to utilize generative models like flow-matching models to improve LLMs. It is also a good example of breaking the typical assumption of the existence of universal intervention vectors, and introducing input-specific intervention vectors.

**Theoretical Claims:**

There is no theorem or theoretical proof in this paper.

---

> ### Author Rebuttal · Authors · 2025-04-01
>
> We thank the reviewer for their valuable feedback and constructive suggestions. We address the questions as follows:
>
> ---
> **Q1: General utility.**
> A1: Thank you for your suggestion. We conduct experiments to test the general utility on MMLU. We evaluate Llama-3-8B-Instruct and TruthFlow on the whole 57 subjects of MMLU in a 5-shot prompt setting. The results presented in the table below indicate that the MMLU accuracy with TruthFlow shows only a minor decrease of 0.2%, which suggests that TruthFlow does not hurt the LLM's general utility while improving factuality.
>
> | Method      |  Acc. (%)     |
> | ----------- |:-------------:|
> | Llama3 Base |     65.77     |
> | TruthFlow   |     65.57     |
>
> **Q2: How to get the blue arrow in Figure 2?**
> A2: We apologize for the confusion. In short, the blue arrow is the average over all light blue arrows. Specifically, we first extract truthful and hallucinated states and reduce the dimensionality. We denote truthful data points as $\{ (x_i^t, y_i^t ) \}_{i=1}^N$
>
> and hallucinated data points as $\{ (x_i^h, y_i^h ) \}_{i=1}^N$
>
> where $N$ refers to the number of data points and $x\in\mathbb{R}, y\in\mathbb{R}$ as the coordinate values of x-axis and y-axis. Then we calculate the mean point $\left(\bar{x}^t, \bar{y}^t\right) = \left(\frac{1}{N}\sum_{i=1}^N x_i^t, \frac{1}{N}\sum_{i=1}^N y_i^t\right)$ and $\left(\bar{x}^h, \bar{y}^h\right) = \left(\frac{1}{N}\sum_{i=1}^N x_i^h, \frac{1}{N}\sum_{i=1}^N y_i^h\right)$. The blue arrow is from the mean hallucinated data point to the mean truthful data point, $\left(\bar{x}^t - \bar{x}^h, \bar{y}^t - \bar{y}^h\right).$
>
> **Q3: Compare with post-training baselines such as GRATH**
> A3: Thank you for your question. Please note that TruthFlow is a representation intervention method, which only involves the inference stage of LLMs, with no additional “post-training” on LLMs. While methods such as GRATH require additional training/finetuning of LLMs which are much more costly compared with inference-based solutions. Therefore, it is unfair to compare with post-training methods if not counting the cost. Furthermore, GRATH actually requires an iterative training schedule with updated training data, which makes it even harder to compare.
> Nevertheless, we still conduct experiments to compare our TruthFlow with GRATH using their publicly released model trained for ten iterative DPO rounds. We report the results in the table below and it achieves very similar performance as TruthFlow. Please note that this particular GRATH model is trained upon ARC-challenge data (much larger than TruthfulQA) and part of TruthfulQA data for over 10 hours, which is much more costly.
>
> | Method      | True  | Info  | True\*Info |
> |:----------- |:-----:|:-----:|:----------:|
> | Llama2 Base | 49.39 | 90.22 |   44.56    |
> | GRATH       | 58.68 | 93.64 |   54.95    |
> | TruthFlow   | 59.41 | 92.42 |   54.91    |
>
> **Q4: Importance of truthful projection. Effect of increasing the number of top singular vectors.**
> A4: Thanks for your questions.
> - We have actually conducted ablation studies on truthful projection in Table 5 of Section 5.4. We found that without projection, TruthFlow still has an overall truthful performance gain over base LLMs. After applying projection, the truthfulness and informativeness are further improved.
> - Figure 3 shows that when the number of top singular vectors is small (e.g., 5), the performance is not good enough due to the severe loss of information. On the other hand, when we choose a larger number of top singular vectors, it gradually becomes sufficient to capture the truthful subspace and thus leads to much more stable performances. At this point, further increasing the number of singular vectors would not help much with the overall performance.
>
> **Q5: Replacing flow matching model with any generative models**
> A5: Thanks for the interesting question. We believe that general diffusion models are not a good choice since they typically start generating from random Gaussian noise (maps from Gaussian distribution to the target distribution). Our TruthFlow leverages the flow matching model to capture the distribution trajectory from the query representation distribution (**which is not Gaussian**) to the correction vector representation distribution. In short, if the generative model can build the trajectory between **any two distributions**, it may work here. If it maps from Gaussian to the target distribution, it is not suited here.

---

> > ### Comment · Reviewer_1fpP · 2025-04-02
> >
> > I thank the authors for the responses, which have addressed my questions, so I raise my score.
> >
> > It would be appreciated if the authors could include A2 and A3 in the paper.

---

> > > ### Author Response · Authors · 2025-04-02
> > >
> > > Thank you for your positive feedback and for raising the score. We sincerely appreciate your recognition of our work. We will include A2 and A3 in our revision.

---

### Official Review · Reviewer_vZHa · 2025-03-16

**Overall Recommendation:** 3

**Summary:**

The paper addresses the hallucination problem in LLMs, where models generate misleading or factually incorrect responses. Unlike prior methods that apply a universal correction vector, TruthFlow employs flow matching to learn query-dependent correction vectors.

**Claims And Evidence:**

The claims in the paper are generally well-supported by experimental evidence. The authors run thorough evaluations on multiple datasets like TruthfulQA and HaluEval, and the results show clear improvements over existing methods. The comparisons with baselines like ITI and TruthX make sense.

However, while the empirical results are convincing, the paper does not provide formal theoretical proofs for some claims, such as the necessity of projecting the correction vector onto a "truthfulness subspace".

**Essential References Not Discussed:**

The paper provides a solid review of related work, covering prior methods in hallucination reduction, representation correction, and flow-based learning. I did not notice any essential references missing.

**Experimental Designs Or Analyses:**

The experimental design is solid, with multiple datasets and strong baseline comparisons. Ablation studies confirm the contributions of  SVD projection.

However, in table 4, would you make any conclusion? which layer should be chosen?

**Methods And Evaluation Criteria:**

The methods and evaluation criteria seem appropriate for the problem. The paper evaluates the approach using TruthfulQA, HaluEval, TriviaQA, and other relevant benchmarks, which makes sense given the goal. Also, both multiple-choice and open-ended evaluation adds robustness to the results.

**Other Comments Or Suggestions:**

Overall, this is a well-executed paper with a good contribution, suggestion see weakness.

**Other Strengths And Weaknesses:**

**Strengths:**

Novelty, flow matching for truthful correction offers a more flexible and effective approach compared to previous static correction methods.

I also appreciate Figure 2, well-designed visualization experiments provide insights into the distributional differences between hallucinated and truthful representations.

The clear experimental setup and good empirical results across multiple benchmarks. The paper is also well-written and easy to follow.

**Weakness:**

It’s unclear how the truthfulness subspace is identified or guaranteed to be accurate, there’s no real proof, just an assumption.

The lack of human evaluation is a limitation, as GPT-4-based assessments, while practical, may not fully capture truthfulness and hallucination reduction.

**Questions For Authors:**

1. How to find the truthfulness subspace, and how to guarantee its correction?

2. The current algorithm and experiments only apply the transition at a specific layer. Is there any conclusion on which layer works best? I’m also curious about the effect of applying the transition across multiple layers simultaneously, would mixing them lead to better results?

3. In Table 1, the Info metric is slightly lower compared to other criteria. I am curious whether this is due to information loss from SVD projection or an effect of flow matching altering the representations. Have you explored this further? Some additional analysis on this trade-off would be helpful.

**Relation To Broader Scientific Literature:**

The paper builds on prior work in LLM truthfulness, hallucination reduction, and representation intervention. It extends methods like ITI and TruthX, which use representation-based corrections, by introducing query-specific Flow Matching for more adaptive interventions. The approach is also related to flow-based generative modeling and low-rank subspace projection (SVD), which have been explored in various contexts but not for truthfulness correction in LLMs. Overall, the contribution is well-positioned within existing literature and provides a novel improvement to LLM alignment.

**Theoretical Claims:**

The theoretical foundation of the paper relies on flow matching and SVD decomposition, both of which are well-established techniques. However, the paper does not provide additional formal theoretical proofs beyond these foundations.

---

> ### Author Rebuttal · Authors · 2025-04-01
>
> We thank the reviewer for their valuable feedback and constructive suggestions. We address the questions as follows:
>
> ---
> **Q1: Lack formal theoretical proofs for some claims, such as the projection onto a "truthfulness subspace"**
>
> A1: First, we would like to emphasize that we do not claim theoretical contributions in this work. This is an empirical work to provide strategies for mitigating hallucinations in LLMs.  But since you asked, we would try our best to show that the idea of the truthful subspace is more than simply our assumption despite that it is hard to formally prove its existence.
>
> Intuitively, the top singular vectors of the matrix correspond to the main basis directions that can point from hallucinated states to truthful states. From this perspective, calling the subspace of these top singular vectors a "truthfulness subspace" is reasonable. Existing works such as [1] also apply similar approaches like SVD to identify "toxic subspace" and justify the identified subspace with theoretical insights. Of course, what is the accurate number of top singular vectors to make this subspace still remains a mystery. We set it as a hyperparameter now but we would like to explore more on this in our future work.
>
> [1] Uppaal et al., Model editing as a robust and denoised variant of DPO: A case study on toxicity. ICLR 2025
>
> **Q2: Conclusion for Table 4? Which layer should be chosen?**
>
> A2: The direct conclusion from Table 4 is that for our method, intervention at the 12th layer is the best compared to the other layers. Note that prior works (such as TruthX [2] and BiPO [3]) all suggest that intervention at the intermediate layers of the model typically leads to the best results. This aligns with our findings in Table 4.
>
> In practice, the choice of intervention layer is more of an empirical hyperparameter and may change due to the LLM architecture, the data, the specific intervention method, etc. However, we notice that the best layer is relatively consistent in models of the same series with similar parameter amounts.
>
> [2] Zhang et al., TruthX: Alleviating Hallucinations by Editing Large Language Models in Truthful Space. ACL 2024
>
> [3] Cao et al., Personalized Steering of Large Language Models: Versatile Steering Vectors Through Bi-directional Preference Optimization. NeurIPS 2024
>
> **Q3: What about the effect of applying the transition across multiple layers simultaneously?**
>
> A3: Thank you for your suggestion. We conduct experiments to apply the transition across two layers simultaneously on Llama-3-8B-Instruct model. Specifically, we extract hidden states from the two selected layers and concat them to form a larger vector to train the flow model. We test some layer combinations on the TruthfulQA open-generation task and report the results in the table below. Applying intervention across two layers simultaneously may slightly (not always) improve the single-layer intervention, but the performance gain is not very significant. We will leave this multi-layer TruthFlow as our future work.
>
> | Layers | True  | Info  | True\*Info |
> | ------ |:-----:|:-----:|:----------:|
> | 12, 13 | 62.10 | 93.40 |   58.00    |
> | 12, 14 | 66.26 | 93.64 |   62.05    |
> | 12, 15 | 64.55 | 93.15 |   60.13    |
> | 12, 20 | 66.50 | 94.13 |   62.60    |
>
> **Q4: The lack of human evaluation.**
>
> A4: Thank you for your suggestion. Certainly, human evaluations would make the results more convincing. Yet our experimental design follows most existing works in this direction and we believe it is enough to justify its better performance. We (the research team) also manually check the generated results and feel consistent with our numerical results. Due to time limitations, we would leave human evaluation as our future work.
>
> **Q5: Reason for slightly lower Info metric.**
>
> A5: We actually observed and analyzed this phenomenon in the paper's “Qualitative Study” paragraph before section 5. Specifically, we find that some of the best answers in TruthfulQA are labeled as less informative, (e.g. the best answer is “I have no comment”). And since TruthFlow makes the model output more truthful, it inevitably causes the info score to be lower on those questions (see an example in Table 2). This leads to the Info score being lower. Please note that TruthFlow can still provide informative answers to other questions whose correct answer is indeed informative.

---

### Official Review · Reviewer_P7qz · 2025-03-17

**Overall Recommendation:** 4

**Summary:**

This paper proposes TruthFlow, a method for improving the truthfulness of LLMs by applying query-specific correction vectors to model representations during inference. Unlike prior work such as ITI that uses a fixed correction vector, TruthFlow employs Flow Matching to generate dynamic interventions tailored to each query. Experiments on TruthfulQA and other benchmarks show that TruthFlow outperforms existing intervention methods and generalizes well across models and datasets.

**Claims And Evidence:**

The paper makes two main claims: (1) that query-specific correction vectors generated via flow matching are more effective than fixed intervention vectors in improving LLM truthfulness, and (2) that TruthFlow outperforms existing intervention methods and generalizes across models and datasets. Overall, the experimental results provide convincing evidence for both claims.

**Essential References Not Discussed:**

The improvements on datasets other than TruthfulQA are not as significant as those on TruthfulQA, likely due to the limited expressive power of the flow-matching model. Liu et al. observed a similar phenomenon and provided an explanation.

Liu et al., Enhancing Language Model Factuality via Activation-Based Confidence Calibration and Guided Decoding. EMNLP 2024

**Experimental Designs Or Analyses:**

Yes. I checked all the given experiments. The experimental designs and analyses are overall sound.

**Methods And Evaluation Criteria:**

The motivation of the proposed method is clear and the evaluation criteria are also appropriate.

**Other Comments Or Suggestions:**

Minor: The writing is generally clear, but adding a summary table comparing TruthFlow with prior intervention methods (e.g., ITI, P-ITI) could help highlight key differences for readers.

**Other Strengths And Weaknesses:**

Strengths:
* The proposed method is well-motivated, which addressed limitations of existing unified-vector methods like ITI.
* The experimental results demonstrate the effectiveness of TruthFlow, the experimental design is reasonable and analysis is solid.
* The paper is clearly written and well-organized

Weakness:
* Theoretical justifications for why flow matching is particularly well-suited for factuality interventions (beyond empirical success) are under-explored.

**Questions For Authors:**

* Have you considered applying TruthFlow to tasks beyond factual QA, such as dialogue or summarization?
* Could you elaborate on how sensitive the method is to hyperparameters like flow model capacity, training data size, and choice of projection subspace?

**Relation To Broader Scientific Literature:**

This paper relates to work on representation interventions for improving factuality in LLMs, such as ITI. Compared to these methods, which apply unified correction vectors, TruthFlow introduces a flexible, query-specific approach via flow matching. It also connects to broader research on controlling LLM behavior through latent space manipulation and builds upon flow-based models used in representation learning. Additionally, the paper aligns with literature on LLM hallucination mitigation and factuality benchmarks like TruthfulQA, contributing a new method that performs well on these established datasets.

**Theoretical Claims:**

The paper does not present formal theoretical proofs but introduces a flow-matching-based framework for learning query-specific intervention vectors. I have checked the conceptual formulation, and the overall idea of using flow matching to transform hallucinated representations toward truthful ones is reasonable and aligned with prior work on flow models.

---

> ### Author Rebuttal · Authors · 2025-04-01
>
> We thank the reviewer for the valuable feedback and constructive suggestions. We address the questions as follows:
>
> ---
>  **Q1: New reference.**
>
> A1: We appreciate the reviewer for pointing out the reference [1]. We will discuss and cite this work in our revision. One thing to clarify is that we test other datasets for transferability while [1] trains their method on various datasets. Therefore, our case is actually more challenging and smaller performance gain under transferability experiment is reasonable.
> [1] Liu et al., Enhancing Language Model Factuality via Activation-Based Confidence Calibration and Guided Decoding. EMNLP 2024
>
> **Q2: Theoretical justifications for why flow matching is suited.**
>
> A2: From our analysis in Sec 3.2, we hope to obtain a query-specific solution that requires us to build a mapping from the query representation distribution to the corresponding correction vector distribution. Furthermore, this solution should also be efficient and well-generalizable, to make it practical to use in LLM inference situations. Flow matching is well-known for building a linear trajectory from source to target distributions, which satisfies our need to capture the distribution mapping. Since the trajectory is linear, it's easier/faster to sample and generalize well. Thus we believe flow matching is well-suited for our goals here. We hope this addresses your concern.
>
> **Q3: Tasks beyond factual QA.**
>
> A3: Note that it is non-typical for factuality-related works to test dialogue and summarization tasks (since these tasks focus more on how LLMs handle and understand context rather than test truthfulness). However, we have tried to train TruthFlow on XSum [2]. Due to time constraints, we randomly sample a subset of data (same size as TruthfulQA) to test its performance. Considering our training process requires pairwise data but there's no "incorrect summary" in XSum, we use GPT-4o to generate seemingly plausible but incorrect summaries for training data. We use the commonly used ROUGE metric for XSum to evaluate. The results suggest that TruthFlow achieves significant improvement upon the base LLM for summarization.
>
> |Method|ROUGE-1|ROUGE-2|ROUGE-L|
> | ----------- |:-------:|:-------:|:-------:|
> | Llama3 base|25.52|7.22 |18.52|
> | TruthFlow|27.41|8.44 |20.14|
>
> In addition, we train TruthFlow on **medical hallucination** benchmark [3] and evaluate following the same set of evaluation metrics (True, Info, and True\*Info scores). The results are presented in the table below. Although slight decrease in Info, TruthFlow outperforms base LLM and ITI in True and True\*Info, demonstrating the effectiveness of TruthFlow in medical domain hallucinations.
>
> | Method      | True  | Info  | True\*Info |
> | ----------- |:-----:|:-----:|:----------:|
> | Llama3 base| 42.54 | 96.82 |   41.19    |
> | ITI |54.77 | 68.70 | 37.63 |
> | TruthFlow   | 57.21 | 94.87 |   54.27    |
>
> [2] Narayan et al., "Don’t give me the details, just the summary." Topic-Aware Convolutional Neural Networks for Extreme Summarization. EMNLP 2018
>
> [3] Pandit et al., MedHallu: A Comprehensive Benchmark for Detecting Medical Hallucinations in Large Language Models. arXiv preprint
>
> **Q4: Hyperparameter sensitivity.**
>
> A4: **Model Capacity.** We conduct an ablation study using three sizes of the model capacity (by adjusting "depth" and "feature_scale"). The size of the small, middle, and large network is 0.05B, 0.11B, and 0.2B in bytes. We test them on TruthfulQA. When the neural network is small, it has difficulty fitting the training data, leading to generating query-specific truthful vectors of lower quality. In comparison, when the model capacity is large enough to fit the training data, the performance becomes stable. The additional parameters do not largely improve TruthFlow-L over TruthFlow-M.
> | Size           | True  | Info  | True\*Info |
> |:-------------- |:-----:| ----- |:----------:|
> | TruthFlow-S    | 63.08 | 88.51 |   55.83    |
> | TruthFlow-M    | 64.79 | 94.38 |   61.15    |
> | TruthFlow-L    | 66.01 | 94.13 |   62.13    |
>
> **Training data size.** We conduct an ablation study using a random subset of TruthfulQA training data. We find that using 1/2 of the original training data leads to worse performance on open-generation task and using 1/4 of training data leads to a more significant performance drop.
>
> | Training data size | True  | Info  | True\*Info |
> |:------------------:|:-----:|:-----:|:----------:|
> | 1/4 of original   | 53.06 | 64.79 |   34.38    |
> |  1/2 of original  | 65.77 | 88.02 |   57.89    |
> |        original    | 64.79 | 94.38 |   61.15    |
>
> **Choice of subspace.** We have already conducted ablation studies on the number of top singular vectors in Figure 3, which determines the subspace. In particular, selecting too many singular vectors generally means retaining most information but could also keep the noisy or hallucinated information while selecting too few singular vectors could lead to severe loss of information.

---

### Decision · Program_Chairs · 2025-05-01

**Decision:**

Accept (poster)

**Comment:**

The paper presents a novel TruthFlow approach that uses flow matching to generate query-specific correction vectors, offering an alternative to static intervention methods for improving LLM truthfulness.
Extensive empirical evaluations across benchmarks like TruthfulQA, HaluEval, and TriviaQA demonstrate measurable improvements over established baselines.
Nonetheless, the theoretical justifications were somewhat under-explored, with some concerns raised about the lack of formal proofs and rigorous statistical validations.
Issues such as inconsistent hyperparameter tuning and limited cross-domain analysis hindered also the overall impact of the contribution.

Since the raised issues were adequately addressed during the rebuttal, an acceptance is recommended, provided that the proposed improvements are incorporated into the paper